# Opto-Microfluidic Fabry-Perot Sensor with Extended Air Cavity and Enhanced Pressure Sensitivity

**DOI:** 10.3390/mi13010019

**Published:** 2021-12-24

**Authors:** Pengfei Zhang, Chao Wang, Liuwei Wan, Qianqian Zhang, Zidan Gong, Zixiong Qin, Chi Chiu Chan

**Affiliations:** 1The Center for Smart Sensing System (S3), Julong College, Shenzhen Technology University, Shenzhen 518118, China; zhangpengfei@stu.gxnu.edu.cn (P.Z.); wanliuwei@sztu.edu.cn (L.W.); zhangqianqian@sztu.edu.cn (Q.Z.); ccchan@sztu.edu.cn (C.C.C.); 2Physical Science and Technology, Guangxi Normal University, Guilin 541004, China; zixiongqin@163.com; 3Sino-German College of Intelligent Manufacturing, Shenzhen Technology University, Shenzhen 518118, China; gongzidan@sztu.edu.cn

**Keywords:** Fabry-Perot Interferometer (FPI), optical fiber sensor, static pressure sensing, micro-fluidic

## Abstract

An opto-microfluidic static pressure sensor based on a fiber Fabry-Perot Interferometer (FPI) with extended air cavity for enhancing the measuring sensitivity is proposed. The FPI is constructed in a microfluidic channel by the combination of the fixed fiber-end reflection and floating liquid surface reflection faces. A change of the aquatic pressure will cause a drift of the liquid surface and the pressure can be measured by detecting the shift of the FPI spectrum. Sensitivity of the sensor structure can be enhanced significantly by extending the air region of the FPI. The structure is manufactured by using a common single-mode optical fiber, and a silica capillary with the inner wall coated with a hydrophobic film. A sample with 3500 μm air cavity length has demonstrated the pressure sensitivity of about 32.4 μm/kPa, and the temperature cross-sensitivity of about 0.33 kPa/K.

## 1. Introduction

Optical fiber pressure sensors are miniature in size, immune to electromagnetic interference and enable remote detection, which is suitable for the applications in biomedical, industrial and environmental safety monitoring [1,2,3]. Among all fiber-type pressure sensors, the Fabry-Perot (FP) interferometric configuration is highly sensitive, compact, and convenient for one-end operation [4]. The measurement schemes of FPI sensors are typically based on two major principles: the change of either in-cavity refractive index (RI) or the cavity-length or both.

The optical fiber FP pressure sensors based on RI changes have no mechanical moving parts in the structure and can be demonstrated by simply connecting a hollow-core fiber between two solid-core fibers. However, their sensitivities were typically 3–5 nm/MPa, which were limited by the small pressure-index coefficient of the gas in the hollow core [5,6,7]. Another type of pressure sensor is based on cavity-length change, where the membrane with large size and small thickness is usually used as the reflective surface of the FP interferometer (FPI) for achieving a higher sensitivity. Membranes of different materials, such as silica [8,9], sapphire [10], diamond [11], silver [12], graphene [13,14] and polymer [15], have been used in building this FPI pressure sensors. Sensors made with silica and sapphire membranes have low sensitivities, however, they can be applied in a high temperature environment above 1000 °C [10]. The sensor with diamond diaphragm can withstand high pressure up to 6.8 MPa [11]. Since graphene and polymer material films have good elasticity and mechanical properties, pressure sensors based on these materials have higher sensitivity. This type of pressure sensor made with a graphene film of ~100 nm thickness and 125 μm diameter has reached a cavity-length sensitivity of 1.1 μm/kPa at 10 kHz [16]. When a Parylene-C diaphragm with a larger diameter of 9 mm and a thickness of 500 nm was used in the sensor, the sensitivity has reached 2.2 × 10^3^ μm/kPa at 20 Hz [15]. The pressure sensors with large area and nano-thickness membrane are fragile to surface defects and environmental influences, hence may not stable and durable enough in static pressure sensing. However, they are useful in the detection of dynamic pressure (acoustic) [13]. On the other hand, the FP pressure sensor based on smaller and thicker membranes can be used for static pressure sensing, but the sensitivity was usually below tens of nm/kPa [12,17]. A compressible micro-FP cavity pressure sensor can simultaneously apply the RI change and the cavity length change to obtain a high cavity-length sensitivity of 6.9 μm/kPa for static pressure [18]. The sensitivity can be further increased to 18.2 μm/kPa by using an anti-resonant hollow-core fiber with selective opening at fiber-end [19].

In this paper, a simple, high-sensitivity optical fiber FP pressure sensor is proposed by using a combination of a common single mode fiber (SMF), a silica capillary tube and a small section of liquid. The new FP sensing configuration has an extended air cavity (EAC) besides the normal FP air cavity, which is the key to sensitivity enhancement. The effects of the structural parameters of the cavity on the measurement performance is analyzed in detail. An optimized sensor sample can reach an extremely high measuring sensitivity of 32.4 μm/kPa.

## 2. Structure and Principle of the Pressure Sensor

The structure of the FP pressure sensor is shown in Figure 1. A flat-cut SMF tip is sealed in a silica capillary at the center position and the fiber-end is close to the capillary opening. When the liquid is purged into the capillary from the open end of the capillary, incident light from the SMF will be reflected, respectively, from the flat fiber-end and the air-liquid surface inside the capillary, hence construct an FPI. In the reflection spectrum, interference fringes of the FPI will shift while the liquid surface move under the pressure change of the liquid. Instant pressure can be measured from the shift of the fringes. Different from the sensors reported in [18], this sensing structure possesses an additional air cavity between the UV-adhesive (ergo 8500) and the fiber-end, which will enhance the measuring sensitivity significantly. The inner wall of the capillary is coated with a hydrophobic material layer (Daikin OPTOOL UD509) to slow down the liquid flow rate inside the capillary.

In the structure, the lengths of the EAC, the initial FPI cavity and instant FPI cavity are defined by *L_e_*, *L*_0_ and *L_x_*, respectively. The reflected light intensity *I* of the FPI can be expressed by: *I* = *I*_1_ + *I*_2_ + *2*(*I*_1_*I*_2_)^1/2^*cos*[*(*4π*nL_x_*/*λ)* + *φ*_0_], where *I*_1_ and *I*_2_ are respectively the reflected light intensity from fiber-end and liquid surface, *n* = 1 is the RI of the air in the cavity; λ is the operating wavelength; *φ_0_* is the initial phase. The fiber is carefully adjusted to be perpendicular to the liquid surface and then fixed by UV-adhesive.

At a constant temperature, the relation between *L_x_* and pressure *P* follows the ideal gas law and can be described by:*L_x_*(*P*) = (*P*_0_*A* − *PD_e_^2^L_e_*)/*PD*^2^(1)
where *P*_0_ is the initial pressure, *D* and *d* are the inner diameter of the capillary and the diameter of the optical fiber, respectively. *D_e_* = (*D*^2^ − *d*^2^)^1/2^ is the effective diameter of the EAC region. *A* = *D_e_*^2^*L_e_* + *D*^2^*L*_0_ is a structure-related parameter to simplify the expression. Then, the cavity length sensitivity of FPI can be deduced by the differentiation of formula (1):(2)$${dL}_{x}/dP=-P_{0}A/P^{2}D^{2}=-P_{0}\frac{\left(1-d^{2}/D^{2}\right)L_{e}+L_{0}}{P^{2}}$$

Equation (2) indicates that *L_x_* would be affected by the dimensions of the FPI cavity and the EAC as well. The sensor with smaller ratio of *d/D*, longer *L*_0_ or *L_e_* would exhibit a larger sensitivity. Since the increasing *L*_0_ and *D* will reduce the FPI contrast and affect the size of the sensor head, respectively, and *d* is always the same for a common fiber, so a larger *L_e_* can enhance the measuring sensitivity.

## 3. Experiments and Discussion

These four samples were built with silica capillary (900 μm inner diameter) and single mode fiber (SMF, 125 μm diameter). The sample-1, -2 and -3 have similar *L*_0_ of ~500 μm, but different EAC lengths *L_e_* of 338 μm, 1516 μm and 3003 μm, respectively. The sample-4 has the same total air-cavity length as sample-1 (*L_e_* + *L*_0_ ≈ 840 μm), but different length ratios (*L_e_:L*_0_) of ~3:7 (*L_e_* = 251 μm*, L*_0_ = 589 μm). Figure 2a is micrograph of the sample-3 which has the longest *L_e_*.

The static pressure sensitivities of the samples are measured with the system shown in Figure 2b. The sample-under-test were sealed in a water-filled Teflon pipe with an inner diameter of 3 mm with AB glue (Kafuter 3-ton clear epoxy adhesive). Pressure was applied from the other end of the pipe by using a pneumatic test pump (Fluke 700PTP-1, Washington, DC, USA) and a digital pressure meter (Fluke 700G07, resolution 10 Pa) was used to monitor the applied pressure for comparison. A semiconductor super-luminescent light emitting diode (SLED, EXALOS EXS210048-02, Schlieren, Switzerland), an optical fiber circulator and an optical spectral analyzer (Yokogawa AQ6370D, Tokyo, Japan) were used to measure the FP interference fringes at different pressure levels. The power of the SLED should be smaller than 1 μW to avoid vaporization of water, otherwise water droplets would be generated inside the cavity and greatly increased the loss of the sample.

The sample was immersed in water and ~10 kPa pressure was applied to the water, a clear liquid surface can be observed inside the capillary, as shown in Figure 3a. Figure 3b shows the spectra of the sample-1 at three different pressure levels. When high pressure was applied to the water, the in-capillary water surface would be closer to the fiber-end, hence more reflected light intensity would be collected by the fiber. The fringe visibility was increased to about 8 dB at a pressure of 70 kPa.

The solid lines in Figure 4 are the simulation results of the relationship between the cavity-length and pressure (*L_x_* − *P*) according to Equation (2) of our FPI pressure sensor samples. For comparison, the pressure sensors without EAC (*L_e_* = 0) are also simulated and represented by the dashed lines. In comparison, the sensors with EAC exhibit a higher cavity-length sensitivity (slope of the lines), however, they have a reduction in operating pressure range. For the sampe-1, -2 and -3 with the same *L*_0_, the sensitivity increases significantly with larger *L_e_*. The sensitivity difference between sample-1 and -4 with the same *L*_0_ + *L*_1_ is not obvious. The dark region in Figure 4a represents the area where the pressure sensor does not work. When there is an instant FPI cavity length *L_x_* ≤ 0, the liquid surface touches the fiber end face and the FPI does not exist; when *L_x_* > ~450 μm, the contrast of the FPI fringe is small because the reflected light intensity from the liquid surface (*I*_2_) is too weak. This region could be reduced by using a fiber-end with collimated lens. The points marked in Figure 4a are the experimental results. The FPI cavity lengths are deduced from the measured free spectral range (FSR) by using the formula: *FSR* = *λ*^2^/2*nL_x_*. Figure 4b shows the experimental data of the sample-3 with the longest *L_e_*. Its cavity-length sensitivity is about 32.4 μm/kPa in the measurement range from 2 kPa to 7 kPa. As we know, this is the highest static cavity-length sensitivity in the reported FPI pressure sensors.

The FSR formula of the FPI with EAC can be rewritten as Equation (3).
*FSR*(*P*) = *λ^2^PD^2^*/*2*(*P*_0_*A* − *PD_e_^2^L_e_*)(3)

Figure 5 show the relationship between FSR and Pressure (FSR~P) of the pressure sensor samples at the wavelength of 1580 nm. The sensor with longer *L_e_* would exhibit larger FSR change with pressure, which might also indicate a larger wavelength-shift sensitivity dλ/dP. The dark region in Figure 5 represents the non-working region in which the liquid surface is too far from the fiber−end.

The pressure sensitivity can also be expressed by the wavelength-shift of the fringe. The dip wavelength of the FP interference fringe is given by: *λ* = 4π*L_x_*/[(2*m* + 1)*π* − *φ*_0_], where *m* is a positive integer. The wavelength-shift sensitivity of the sensor can be calculated by substituting the λ into Equation (1) and take the derivative as shown in Equation (4).
(4)$$d\lambda /dP=-\frac{P_{0}A\lambda }{P\left(P_{0}A-PD_{e}{}^{2}L_{e}\right)}$$

Figure 6a shows the simulation results of Equation (4) (solid lines) and experimental results (points) for different samples. The dashed line in the figure is the change of the wavelength of FSR due to the applied pressure of the sensor without EAC (*L_e_* = 0), which is independent of the cavity length *L*_0_ [18]. For our samples, the longer EAC ones would exhibit higher wavelength-shift sensitivity but with smaller measurement range. The wavelength-shifts of sample-3 and -4 were measured by observing a dip wavelength around 1550 nm and within the pressure range of ~1 kPa. The experimental results are given in Figure 6b. The wavelength-shift sensitivity of sample-3 at pressure ~4 kPa is 138.9 nm/kPa and the sensitivity of sample-4 at pressure ~218 kPa is 76.0 nm/kPa. Since the wavelength shift caused by air, the RI variation with the pressure can be estimated to be ~4.2 pm/kPa [5], the sensitivity of the samples mainly comes from the changes of cavity lengths.

Temperature cross-sensitivity of the FPI pressure sensor (sample-3) is measured with a ceramic heating platform (CORNING PC-420D, New York, NY, USA) and a digital thermometer (MITIR, resolution 0.1 K, Zhejiang, China) at a constant pressure of 10 kPa. The temperature range for the test is from 293.15 K to 303.15 K. Figure 7 shows the experimental results, which indicate that the temperature-induced cavity-length sensitivity is 10.7 μm/K. Hence, the temperature cross-sensitivity can be calculated to be about 0.33 kPa/K. The cross-sensitivity of the sample is proportional to the pressure under test and inversely proportional to the environmental temperature [18], and about 40% lower than that of a silver diaphragm-based FPI sensor [12]. This temperature crosstalk could be compensated by the introduction of a temperature sensor, such as a fiber Bragg grating, into the sensor head.

The repeatability of the pressure sensor is also tested in the pressure range of 35 kPa to 35.5 kPa. A sample (*L_e_* 1011 μm, *L*_0_ 507 μm) is tested for increasing pressure at intervals of 0.1 kPa. At each pressure, there are 20 measurements with 2 s intervals. The dip wavelengths of the sample are recorded and represented by box plots in Figure 8. This sample is then tested for decreasing pressure at the same intervals. The data (red dot in Figure 8) of the decreasing pressure process are located near the center of the box. The 25–75% box range is 2 nm maximum at 35.4 kPa measurement, and 1.1 nm minimum at 35.2 kPa measurements. So, the worst measurement error of the sample is about 18 Pa.

## 4. Discussion and Conclusions

This paper introduces a high measurement-sensitivity optical static-pressure sensor. The main contribution to the high sensitivity is by the introduction of an extended air-cavity (EAC) in the normal Fabry-Perot sensing structure. Longer EACs increase the pressure measurement sensitivity of the sensor significantly within a smaller dynamic range. The range could be extended by using beam-collimated fiber-end face in the structure. Sample with ~3 mm EAC exhibits cavity-length sensitivity of ~32.4 μm/kPa within the pressure range from 2 to 7 kPa, and a wavelength shift sensitivity of 138.9 nm/kPa. Temperature cross-sensitivity of the sensor about 10.7 μm/K is measured. This compact and highly sensitive sensing structure can be applied in opto-microfluidic system for static pressure sensing and could also be a good candidate for dynamic pressure sensing.

## Figures and Tables

**Figure 1 micromachines-13-00019-f001:**
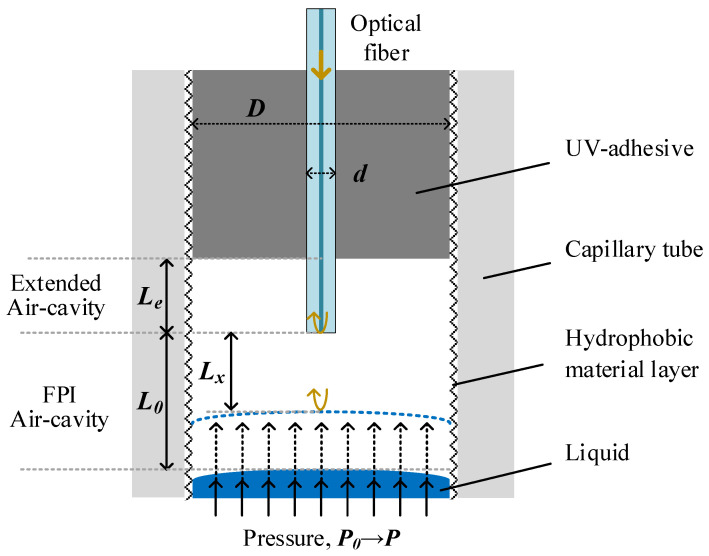
Schematic of pressure sensor with extended air cavity.

**Figure 2 micromachines-13-00019-f002:**
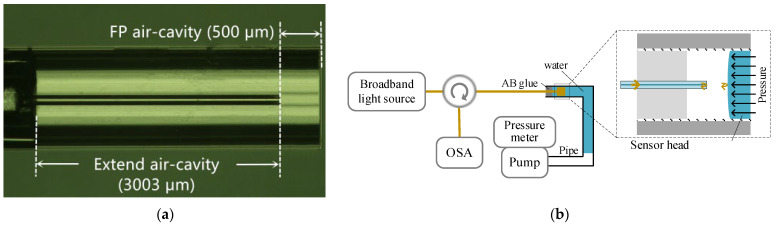
(**a**) Micrograph of sample-4 in air. (**b**) Experimental set-up to test the sensitivity of the pressure sensor head with EAC.

**Figure 3 micromachines-13-00019-f003:**
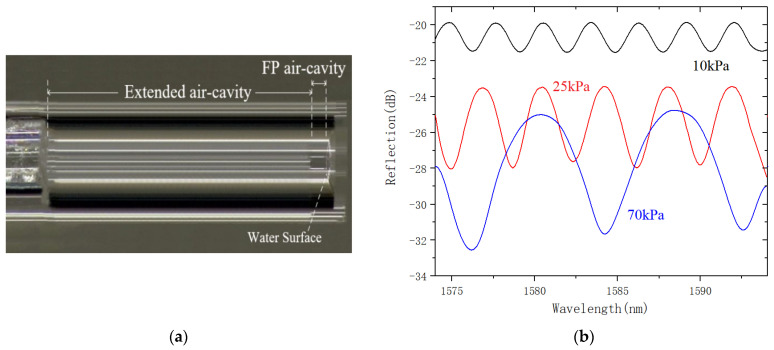
(**a**) Micrograph of sample-3 in water. (**b**) Reflection spectrum of the sensor at different pressures.

**Figure 4 micromachines-13-00019-f004:**
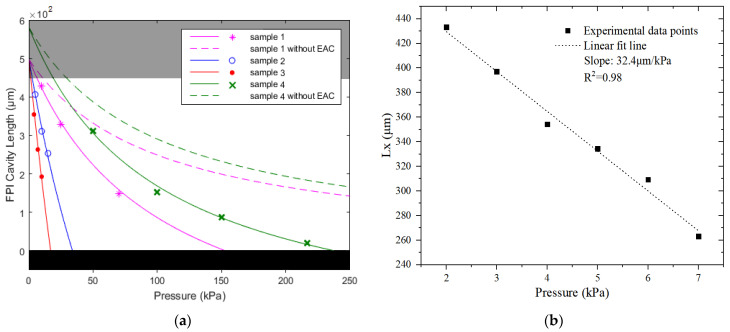
(**a**) Simulation results of the L_x_−P relation (lines) and experimental results (points). (**b**) Experimental results of sample-3.

**Figure 5 micromachines-13-00019-f005:**
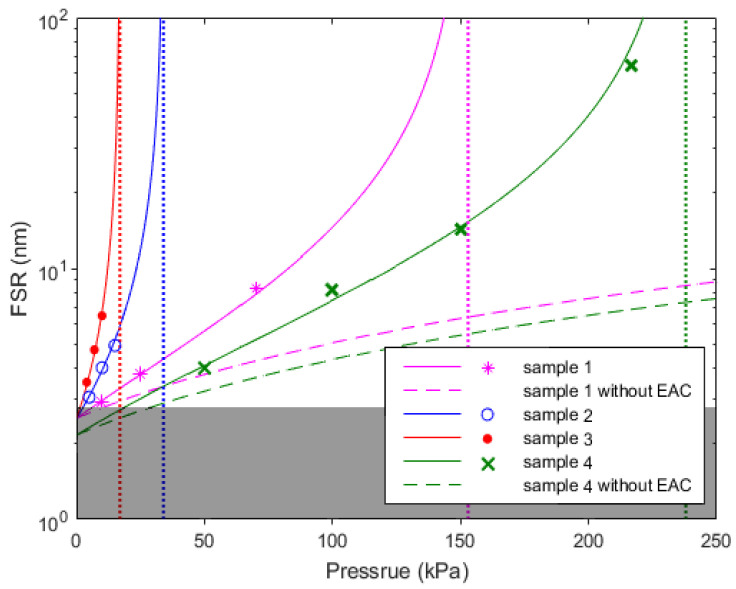
Simulation and experimental results of *FSR* of the sensor samples.

**Figure 6 micromachines-13-00019-f006:**
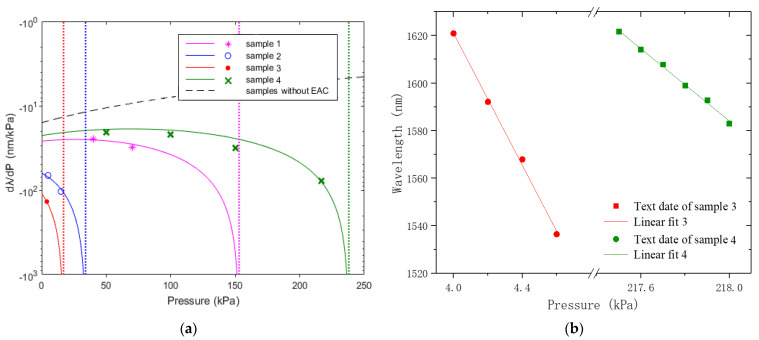
(**a**) Simulation and experimental results of *dλ*/*dP* at different pressure levels; (**b**) wavelength-shift sensitivities of sample-3 and sample-4.

**Figure 7 micromachines-13-00019-f007:**
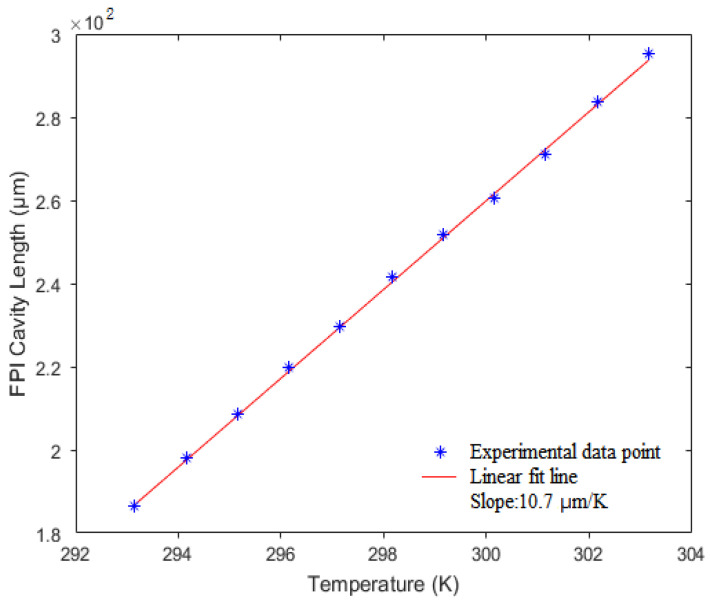
Measured cavity-length of FPI at different temperature with a constant pressure of 10 kPa.

**Figure 8 micromachines-13-00019-f008:**
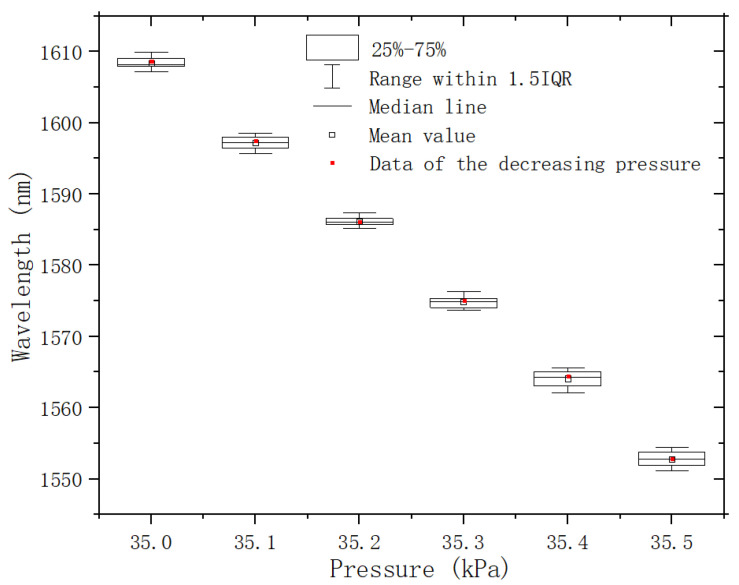
Repeatability test of the pressure sensor.

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
