# Peer review of "Opto-Microfluidic Fabry-Perot Sensor with Extended Air Cavity and Enhanced Pressure Sensitivity"

_micromachines, 2021, doi:10.3390/mi13010019_

Round 1

Reviewer 1 Report

In the manuscript of micromachines-1520157, an opto-microfluidic static pressure sensor based on a fiber Fabry-Perot interferometer (FPI) with extended air cavity for enhancing the measuring sensitivity is proposed. With a microfluidic channel by the combination of the fixed fiber-end reflection and floating liquid surface reflection faces, the sensitivity of the sensor structure can be enhanced significantly by extending the air region of the FPI. The research can give a practical solution with good performance and the manuscript is well written. However, I will provide some comments for enhancing the quality of the manuscript.

  1. To provide the detailed structural and material information of the opto-microfluidic static pressure sensor, including the manufacturer and type of UV-adhesive, hydrophobic material layer and AB glue.
  2. To give the dynamic changing curvature of liquid surface in cavity and discuss the influence of gravity to the shape of liquid surface.
  3. To discuss the influence of temperature.
  4. To provide the overall loss of this system.
  5. To further clarify the innovation in the section of introduction and add more references
  6. To provide the measuring error and repeatability performance of the sensor.

Reviewer 2 Report

In this work, the authors reported the high sensitive pressure sensor using the new FP configuration with extended air cavity (EAC). This paper can give a practical solution for static pressure sensing. Overall, I think the manuscript is well written and the results are interesting. However, I have some comments that require the authors’ attention and call for modification of the manuscript.

  1. What is the overall loss of this system? I think there are two major losses in the proposed scheme. The first one is the absorption in water, water has the high absorption at the ~1.5 um range, which is the operating wavelength in this experiment. The second one is coupling loss while light propagates the optical fiber into free space and return to the optical fiber. I wonder the overall loss in the proposed scheme considering all losses.
  2. It is not clear why the sensitivity is dependent on the EAC at least for me. I think explaining Eq. (1) in detail could be helpful for this. Also, Eq. (1) needs references. I recommend that it is helpful for readers to understand this paper by adding an explanation of the operation principle of the sensitivity enhancement by the EAC.
  3. In line 183, the authors expected that their scheme can also be applied for dynamic pressure sensing. However, the OSA is used in the proposed system and it always require a specific time for scanning the wavelength. Do the authors have any solution for this problem in mind?

Round 2

Reviewer 1 Report

All the comments and suggestions are addressed, and I recommend the manuscript to be accepted in present form.